# Genomic diversity of bacteriophages infecting *Rhodobacter capsulatus* and their relatedness to its gene transfer agent RcGTA

Jackson Rapala[1], Brenda Miller[1], Maximiliano Garcia[1], Megan Dolan[1], Matthew Bockman[1], Mats Hansson[2], Daniel A. Russell[3], Rebecca A. Garlena[3], Steven G. Cresawn[4], Alexander B. Westbye[5¤], J. Thomas Beatty[5], Richard M. Alvey[1]*, David W. Bollivar[1]

1 Department of Biology, Illinois Wesleyan University, Bloomington, Illinois, United States of America,
2 Department of Biology, Lund University, Sölvegatan 35B, Lund, Sweden, 3 Department of Biological Sciences, University of Pittsburgh, Pittsburgh, PA, United States of America, 4 Department of Biology, James Madison University, Harrisonburg, Virginia, United States of America, 5 Department of Microbiology and Immunology, University of British Columbia, Vancouver, Canada

¤ Current address: Department of Medical Biochemistry, Hormone Laboratory, Oslo University Hospital, Oslo, Norway
* ralvey@iwu.edu

**Data Availability Statement:** All 26 files are available from the NCBI GenBank database (accession number(s) NC_042049, NC_028954,

## Abstract

The diversity of bacteriophages is likely unparalleled in the biome due to the immense variety of hosts and the multitude of viruses that infect them. Recent efforts have led to description at the genomic level of numerous bacteriophages that infect the Actinobacteria, but relatively little is known about those infecting other prokaryotic phyla, such as the purple non-sulfur photosynthetic α-proteobacterium *Rhodobacter capsulatus*. This species is a common inhabitant of freshwater ecosystems and has been an important model system for the study of photosynthesis. Additionally, it is notable for its utilization of a unique form of horizontal gene transfer via a bacteriophage-like element known as the gene transfer agent (RcGTA). Only three bacteriophages of *R. capsulatus* had been sequenced prior to this report. Isolation and characterization at the genomic level of 26 new bacteriophages infecting this host advances the understanding of bacteriophage diversity and the origins of RcGTA. These newly discovered isolates can be grouped along with three that were previously sequenced to form six clusters with four remaining as single representatives. These bacteriophages share genes with RcGTA that seem to be related to host recognition. One isolate was found to cause lysis of a marine bacterium when exposed to high-titer lysate. Although some clusters are more highly represented in the sequenced genomes, it is evident that many more bacteriophage types that infect *R. capsulatus* are likely to be found in the future.

KT_253150, NC_029097, NC_041963, MW677509, MW677510, MW677511, MW677512, MW677513, MW677514, MW677515, MW677516, MW677517, MW677518, MW677519, MW677520, MW677521, MW677522, MW677523, MW677524, MW677525, MW677526, MW677527, MW677528, MW677529.)

**Funding:** The research of JTB was supported by a grant (RGPIN 2018-03898) from the Canadian Natural Sciences and Engineering Research Council (NSERC). The research of DWB was supported by funds from the Illinois Wesleyan University Miner Linnaeus Sherff endowed professorship in Botany. The funders had no role in study design, data collection and analysis, decision to publish, or preparation of the manuscript.

**Competing interests:** The authors have declared that no competing interests exist.

## Introduction

Bacteriophages (phages) are the most massively abundant and diverse biological entities with an estimated $10^{31}$ particles in the biosphere [1, 2]. They are known to greatly impact microbial populations in a variety of ways including the virulence and persistence of bacterial pathogens [3]. Concerted efforts to identify phages of Actinobacteria such as mycobacteria, *Arthrobacter*, *Gordonia*, *Microbacterium*, *Rhodococcus*, *Streptomyces* as well as studies of enterobacteria, *Bacillus*, and *Pseudomonas* have begun to fill in the missing information about these neglected but impactful entities [4–11]. In terms of α-proteobacteria, phages that infect the hosts *Caulobacter*, *Ruegeria*, and *Dinoroseobacter* have been identified and sequenced [12, 13]. To date however, only a few phages of *Rhodobacter capsulatus* have been examined in significant detail [14–16].

*R. capsulatus* is a photosynthetic α-proteobacterium with the ability to grow under a wide variety of conditions and has been used as a model for photosynthesis and nitrogen fixation. Part of the reason that *R. capsulatus* was developed as a model system was the presence of a genetic system that allowed for simple transduction-like gene transfer and generation of site-directed gene knockouts. This system was based on a phage-like entity known as a gene transfer agent (RcGTA). RcGTA has been studied due to the implications it has for the transfer of genetic information between related bacteria in the environment [17]. The majority of the structural proteins for RcGTA are encoded in an approximately 14 kb region of the *R. capsulatus* genome with additional genes (eg. for fibers, lytic release, and regulation) found elsewhere in the genome [18, 19]. In contrast to phages, individual particles can only package around 4 kb [20]. This results in the inability of RcGTA to package its full genome and instead particles contain random fragments of host DNA, though there is some evidence that this is not completely random [21]. Similar systems have since been identified in a variety of other bacterial species [17].

Until recently, phages that infect *R. capsulatus* have received much less attention than those of other bacteria. In the mid-1970s, Wall et al. [14] and Schmidt et al. [22] isolated nearly 100 phages of *R. capsulatus* from sewage and characterized them based on host range. One of these, RC1, was selected for further study. This work established host range and potential effects of RC1 on RcGTA production and suggested that presence of RC1 was bioenergetically costly to host cells. It also placed RC1 in the *Siphoviridae* based on morphology [22]. None of the phages from these studies have been characterized at the genomic level. A sequenced genome for a *R. capsulatus* strain E32 phage also named "RC1" has been deposited in GenBank (accession number JF974308). This isolate (not the same RC1 as Wall et al.) was obtained through prophage induction of gas hydrate sediment samples from the Pacific Ocean [23]. It shares some sequence similarity with Mu-type phages that infect *Escherichia coli*. Since then, two additional phages of *R. capsulatus* have been identified as prophages present in strains SB1003 and Y262. One, RcapMu, could be induced to excise and reinfect, and as the name suggests, is also a relative of Mu transposing phages [15]. The other, RcapNL, was isolated and sequenced, but was not further characterized in depth because no other host has been identified that it can infect [16].

The work presented here describes an important expansion in the number of sequenced *R. capsulatus* phages. Together with those previously sequenced, these 26 new phages can be organized into six clusters (designated RcA, RcB, RcC, RcD, RcE, and RcF) with at least two members and four additional singleton groups represented by a single member. Phages that infect *R. capsulatus* are highly diverse, sharing only limited gene conservation between clusters. The presence of shared genes between RcGTA, and some of the phages is likely connected to host recognition based on the position of these gene products in the newly determined

structure of RcGTA [18]. It is also intriguing to note that one of these phages with shared proteins, is able to induce cell lysis of the marine bacterium *Dinoroseobacter shibae*, suggesting *R. capsulatus* phages might serve as an interesting model for examining host range evolution.

## Results

### *R. capsulatus* phage isolation and morphology

26 novel phages infecting the bacterium *R. capsulatus* were isolated from collected water samples. All but one of these were isolated using the host strain YW1 C6; the exception being RcSimone-Håstad isolated on strain SB1003. Most were isolated from samples collected in the USA with the notable exceptions of RcSimone-Håstad that was isolated from a sample collected in Håstad, Sweden and RcThunderbird isolated from a sample collected near a wastewater treatment plant in Vancouver, British Columbia in Canada. All of the phages formed plaques on host lawns.

Each of the phages described here has a dsDNA genome and a flexible noncontractile tail placing it in the *Siphoviridae* branch of the *Caudovirales* (Fig 1). All have isometric capsids with the exception of RcSimone-Håstad, which has a prolate capsid. Measured capsid diameters and tail lengths which were found to be similar for clustered phages are summarized in Table 1.

### Host range/plaque formation

The ability of these new isolates to form plaques on the following strains of *R. capsulatus* were examined: YW1, YW2, B6, B10, St. Louis, 37B4, and Iona (Table 2). These strains were chosen because they represent a spectrum of RcGTA interactions, in terms of either production or reception, and all except for Iona and St. Louis have a sequenced genome available [24, 25]. Additionally, two more distantly related marine bacterial species, *Dinoroseobacter shibae* DFL12 and *Ruegeria pomeroyi* DSS3, were also examined as potential hosts. For most of the phages tested, unique patterns of plaque formation allowed for tentative grouping or cluster assignment. With subsequent genomic sequencing these assignments were largely found to be consistent for all the phages of a cluster and the patterns of plaque formation with these host strains were distinct characteristics of particular clusters. The exception to this was RcOceanus which is unique in its cluster due to its inability to infect B10 or St. Louis. The singleton RcZahn was also notable for having the ability to form plaques on *R. capsulatus* strain 37B4 and on *D. shibae*, two hosts that none of the other phage isolates could form plaques on. Additionally, none of the phages tested could form plaques on *R. capsulatus* strain Iona or on *R. pomeroyi*.

### Genometrics

The genomic sequences of 26 phages were determined and used in comparative analyses along with the 14,087 bp region of the RcGTA genome which encodes 17 genes—most of which comprise its structural components—and three previously sequenced phages, (RC1, RcapMu, and RcapNL) (Table 3). Genome sizes for the new isolates range from 35,985 bp with 45 predicted genes (RcCronus) to 101,599 bp with 147 genes (RcZahn). All have GC percentages lower than the host (66.5%) with a range from 54.8% (RcThunderbird) to 65.4% (RcRhea) [24, 25]. The majority of the isolated phage DNAs have defined ends with short 11–13 bp 5'overhangs, but other end types were observed including circularly permuted genomes, direct terminal repeats, and P1 type headful packaging (Table 3).

Our analysis identified 2,350 genes that can be organized into 833 distinct gene families (phams) among the 29 phage genomes and RcGTA structural gene cluster. Of these phams, 367 (44%) were found to be orphams, or genes found in only one phage in this database. There are 5 genes shared by as many as 12 entries. The average gene length is 646 bp with the largest

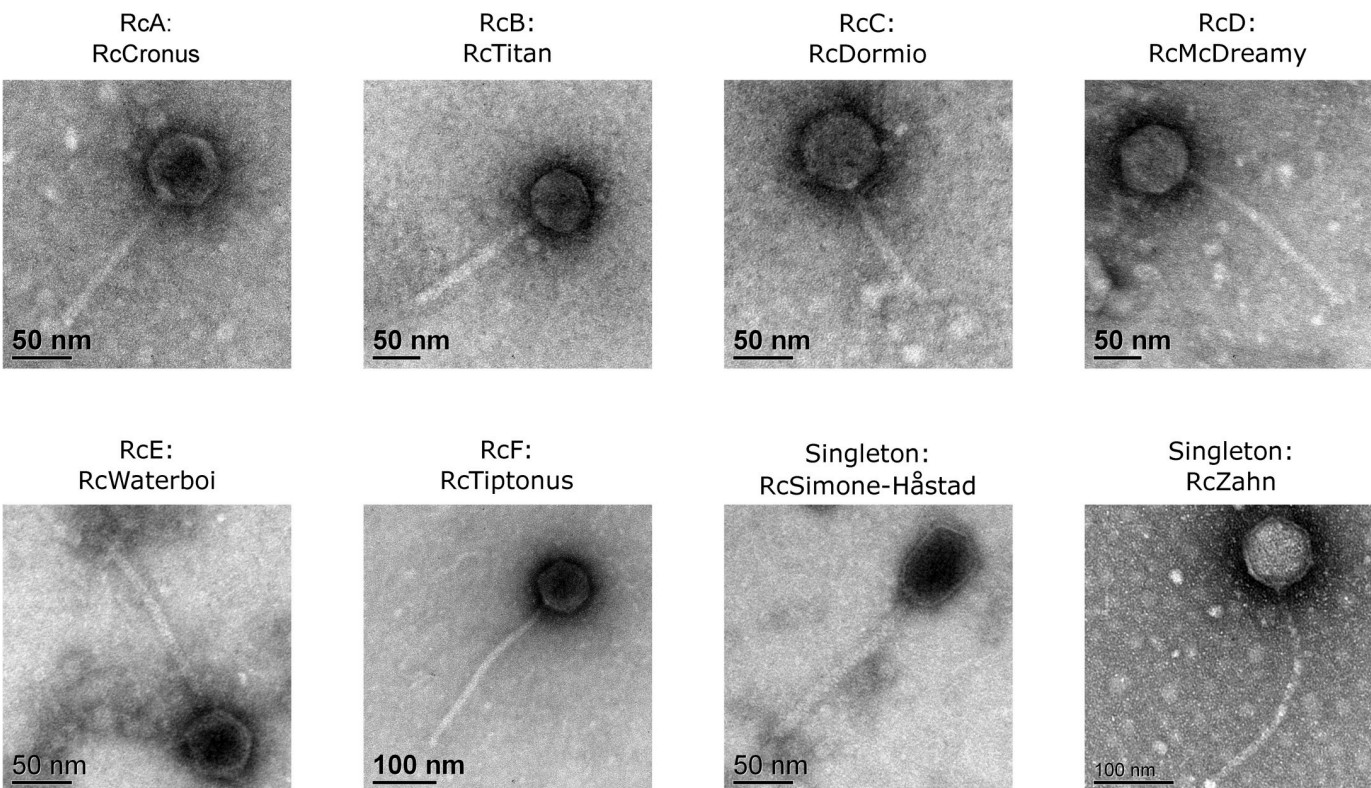

**Fig 1. *R. capsulatus* phage virion morphologies.** Representative transmission electron micrographs of virion particles from each *R. capsulatus* phage cluster shows the presence of *Siphoviridae* morphologies. Cluster designations, RcA-RcF or Singleton, are indicated above the representative phage name.

predicted genes being for the tape measure proteins of cluster RcF phages, RcDurkin and RcTiptonus, (gene #44 in both) which are 5,364 bp.

## Major capsid protein and large terminase subunit comparisons

The major capsid and large terminase subunit protein sequences are commonly used markers for understanding phage phylogeny. Protein-protein BLAST (BLASTP) queries of the

**Table 1. *R. capsulatus* phage virion measurements.**

| Cluster/Phage | Average capsid diameter (nm) | Average tail length (nm) | Number of virions measured |
|---|---|---|---|
| RcA | 59.7 (3.2) | 132.9 (4.6) | 10 |
| RcB | 61.0 (1.8) | 150.2 (6.0) | 21 |
| RcC | 66.2 (2.3) | 115.8 (8.5) | 15 |
| RcD | 73.3 (3.1) | 204.4 (9.7) | 10 |
| RcE | 65.9 (3.1) | 135.8 (18.6) | 6 |
| RcF | 80.6 (3.2) | 297.2 (36.4) | 10 |
| RcSimone-Håstad | 76.7 (4.6) x 54.1 (3.1) | 184.2 (12.8) | 10 |
| RcZahn | 87.1 (4.2) | 296.9 (18.7) | 4 |

Measurements for the capsid diameters and tail lengths represent averages calculated using multiple separate phage particles for multiple representatives of a cluster using ImageJ. For the singletons, RcSimone-Håstad and RcZahn, measurements represent the averages from several independent phage particles. Standard deviations for each average measurement is provided in parentheses.

**Table 2. Plaque formation using spot testing on various hosts.**

| Cluster/ Phage | Host Strain | | | | | | | | |
|---|---|---|---|---|---|---|---|---|---|
| | YW1 | YW2 | B6 | B10 | St. Louis | 37B4 | Iona | *D. shibae* | *R. pomeroyi* |
| RcA | + | - | - | - | - | - | - | - | - |
| RcB | + | +[1] | +[1] | + | + | - | - | - | - |
| RcC | + | - | - | +[2] | +[1,2] | - | - | - | - |
| RcD | + | +[1] | +[1] | +[1] | +[1] | - | - | - | - |
| RcE | + | + | - | - | +[1] | - | - | - | - |
| RcF | + | + | + | + | + | - | - | - | - |
| RcSimone- Håstad | + | - | - | + | +[1] | - | - | - | - |
| RcZahn | + | - | + | - | - | + | - | + | - |

1- Plaques were turbid or much more difficult to discern than on YW1

2 –RcOceanus differed from the other RcC phages in that it was unable to form plaques on B10 or St. Louis.

NCBI Non-redundant protein sequences viruses database (taxid:10239) were used to identify phages with similar capsid and terminase sequences to those of each of the 26 newly and 3 previously sequenced phages. As these sequences tend to be highly conserved between highly related phages the results of this analysis are organized by cluster designation (Table 4). In all instances, except for the result for the major capsid protein of RcapNL, matches were limited to those of phages infecting α- (more frequently) or γ- (less frequently) proteobacteria.

## Phage clustering

The comparison of phage genomes is facilitated by using the phamily designations generated within the Phamerator program to map the shared amino acid coding sequences between phages. Using this information, a visual representation of the shared gene network using Splitstree [26] can be constructed (Fig 2). This analysis reveals six clusters of phages (designated RcA to RcF) with varying numbers of members. Four phages only share between 1 (RcapNL) and 17 (RcZahn) genes with any others in this collection (Table 5) as determined by Phamerator (with its default cutoff of 32.5% CLUSTALW identity) and are described as singletons following the terminology of Hatfull et al [27]. It also should be noted that two clusters have also been described as Genera; RcA is the genus *Cronusvirus*, and RcB is the genus *Titanvirus* [28–30]. Phamerator can also be used to visualize differences in genomes between cluster members (Fig 3). This analysis readily reveals the high level of gene conservation between members of the same cluster (S2–S9 Figs) and that the majority of orphams in this collection are found in the singleton genomes.

Shared protein coding regions do not necessarily indicate shared nucleotide sequence however. Some clusters have members with relatively large differences in the average nucleotide identities with other cluster members. RcKemmy for instance, shares between 85–86% average nucleotide identity (ANI) with any other member of the RcC cluster. Other clusters, such as the RcA cluster, have high nucleotide conservation between members with 98–99% ANI between any pairing of these phages. Comparison of nucleotide sequence conservation amongst phage clusters demonstrates significant variation with occasional pockets of shared sequence. This can be visualized with a dotplot comparison of the catenated genomes with themselves (S1 Fig). The central diagonal indicates self-alignment, but it is also evident that there are genomes that share substantial sequence similarity correlating with the six clusters described above. RcGTA and the four singletons (RC1, RcapNL, RcSimone-Håstad, and

**Table 3. Genometrics of *R. capsulatus* phages and RcGTA.**

| Phage name | Year | Cluster | Type of end | Host[1] | Length[2] | Fold Coverage | GC% | ORFs | lytic/temp[3] | Accession #[4] | Reference |
|---|---|---|---|---|---|---|---|---|---|---|---|
| RcCronus | 2013 | RcA | 5' overhang 13 base | YW1 | 35985 | 1130 | 65.4 | 45 | temperate | NC_042049 | This paper, Bollivar et al |
| RcRhea | 2013 | RcA | 5' overhang 13 base | YW1 | 36065 | 1674 | 65.4 | 45 | temperate | NC_028954 | This paper, Bollivar et al |
| RcSaxon | 2012 | RcA | 5' overhang 13 base | YW1 | 36081 | 250 | 65.4 | 46 | temperate | KT 253150 | This paper, Bollivar et al |
| RcTitan | 2012 | RcB | circularly permuted | YW1 | 44496 | 2088 | 55.1 | 61 | lytic | NC_029097 | This paper, Bollivar et al |
| RcSpartan | 2012 | RcB | circularly permuted | YW1 | 44194 | 151 | 54.9 | 61 | lytic | NC_041963 | This paper, Bollivar et al |
| RcThunderbird | 2015 | RcB | circularly permuted | YW1 | 43941 | 2389 | 54.8 | 61 | lytic | MW677526 | This paper |
| RcHartney | 2018 | RcB | circularly permuted | YW1 | 43528 | 6112 | 55.1 | 60 | lytic | MW677514 | This paper |
| RcOceanus | 2013 | RcC | 5' overhang 11 base | YW1 | 37609 | 425 | 64.2 | 57 | ND[5] | MW677520 | This paper |
| RcDormio | 2015 | RcC | 5' overhang 11 base | YW1 | 41640 | 3975 | 64.1 | 69 | ND[5] | MW677510 | This paper |
| RcBaka | 2016 | RcC | 5' overhang 11 base | YW1 | 41643 | 640 | 64.1 | 70 | ND[5] | MW677509 | This paper |
| RcFrancesLouise | 2016 | RcC | 5' overhang 11 base | YW1 | 42073 | 3354 | 64.0 | 71 | ND[5] | MW677512 | This paper |
| RcHotPocket | 2016 | RcC | 5' overhang 11 base | YW1 | 41765 | 86 | 64.1 | 70 | ND[5] | MW677515 | This paper |
| RcKemmy | 2018 | RcC | 5' overhang 11 base | YW1 | 41345 | 4554 | 63.7 | 69 | ND[5] | MW677517 | This paper |
| RcGingersnap | 2016 | RcD | 5' overhang 12 base | YW1 | 68225 | 3720 | 60.2 | 101 | ND[5] | MW677513 | This paper |
| RcIroh | 2016 | RcD | 5' overhang 12 base | YW1 | 68575 | 2123 | 60.2 | 100 | ND[5] | MW677516 | This paper |
| RcMcDreamy | 2015 | RcD | 5' overhang 12 base | YW1 | 68228 | 3313 | 60 | 101 | ND[5] | MW677518 | This paper |
| RcMrWorf | 2016 | RcD | 5' overhang 12 base | YW1 | 67196 | 5125 | 60 | 99 | ND[5] | MW677519 | This paper |
| RcPutin | 2016 | RcD | 5' overhang 12 base | YW1 | 67605 | 3270 | 60.3 | 100 | ND[5] | MW677522 | This paper |
| RcPescado | 2016 | RcD | 5' overhang 12 base | YW1 | 67494 | 1964 | 60.4 | 99 | ND[5] | MW677521 | This paper |
| RcRios | 2017 | RcD | 5' overhang 12 base | YW1 | 68774 | 1573 | 60.3 | 103 | ND[5] | MW677523 | This paper |
| RcSalem | 2017 | RcD | 5' overhang 12 base | YW1 | 67698 | 1245 | 60 | 101 | ND[5] | MW677524 | This paper |
| RcapMu | 2011 | RcE | Mu-type | SB1003 | 39283 | ND | 64.9 | 59 | temperate | NC_016165 | Fogg et al |
| RcWaterboi | 2016 | RcE | Mu-type | YW1 | 38301 | 6667 | 64.8 | 56 | temperate | MW677528 | This paper |
| RcTiptonus | 2015 | RcF | P1 headful | YW1 | 94091 | 1938 | 57.9 | 139 | ND[5] | MW677527 | This paper |
| RcDurkin | 2018 | RcF | P1 headful | YW1 | 94639 | 252 | 57.8 | 141 | ND[5] | MW677511 | This paper |
| RC1 | 2002 | singleton | Mu-type | E32 | 39573 | ND | 62.3 | 56 | ND[5] | JF974308 | Engelhardt et al. |
| RcSimone-Håstad | 2017 | singleton | 77 base terminal repeat | SB1003 | 63102 | 1267 | 60.7 | 80 | ND[5] | MW677525 | This paper |
| RcZahn | 2018 | singleton | circularly permuted | YW1 | 101599 | 1256 | 60.7 | 147 | ND[5] | MW677529 | This paper |
| RcapNL | 2011 | singleton | circularly permuted | SB1003 | 40489 | ND | 65.1 | 65 | temperate | JQ066768 | Hynes, AP |
| RcGTA | 1974 | N/A | N/A | N/A | 14087 | ND | 69.2 | 17 | N/A | AF181080 | Marrs, B. |

[1]Host strain used for isolation.

[2]Genome length in base pairs.

[3]Lytic or temperate life style, as predicted bioinformatically.

[4]GenBank Accession number.

[5]ND = Not determined.

RcZahn) share little nucleotide sequence identity with the other groups, though there is a small region (~750 bp) of RcSimone-Håstad that is similar to members of the RcC cluster.

Genome BLAST distance phylogeny has been proposed as a robust method for determining phage relationships and proposing genera [32, 33]. The genomes of the 29 *R. capsulatus* phages and the RcGTA structural gene region of the *R. capsulatus* genome were submitted to the VICTOR analysis page at the German Collection of Microorganisms and Cell Cultures (Deutsche

**Table 4. Major capsid protein and large terminase subunit Genbank closest matches.**

| Cluster | Best phage match | | | | | |
| --- | --- | --- | --- | --- | --- | --- |
| | Major capsid protein | % identity | % coverage | Large terminase | % identity | % coverage |
| RcA | Dinoroseobacter phage vB_DshS-R4C | 45.8 | 99 | Dinoroseobacter phage vB_DshS-R4C | 55 | 89 |
| RcB | Pseudomonas phage vB_PaeS_C1 | 74.8 | 98 | Escherichia phage Halfdan | 60.7 | 98 |
| RcC | Stenotrophomonas phage S1 | 33.9 | 99 | Wolbachia phage WOVitA1 | 49.6 | 91 |
| RcD | Ruegeria phage vB_RpoS-V18 | 35.2 | 98 | Loktanella phage pCB2051-A | 44.4 | 97 |
| RcE | Rhizobium phage RR1-B | 73.6 | 99 | Rhizobium phage RR1-B | 72.4 | 99 |
| RcF | Rhizobium phage RHph_I4 | 41.1 | 98 | Stenotrophomonas phage vB_SmaS_DLP_5 | 40 | 48 |
| RcSimone- Håstad | Pseudomonas virus Yua | 71.9 | 100 | Pseudomonas phage PaMx28 | 61 | 98 |
| RcZahn | Rhizobium phage RHph_TM16 | 67 | 99 | Stenotrophomonas phage vB_SmaS_DLP_3 | 53.7 | 83 |
| RcapNL | Burkholderia phage phi6442 | 41.8 | 99 | Paracoccus phage vB_PthS_Pthi1 | 61.5 | 98 |
| RC1 | Rhodovulum phage RS1 | 100 | 100 | Rhodovulum phage RS1 | 100 | 100 |

Sammlung von Mikroorganismen und Zellkulturen, DSMZ) website [32] to determine how this analysis would compare with the Splitstree clustering method (Fig 4). As expected, the

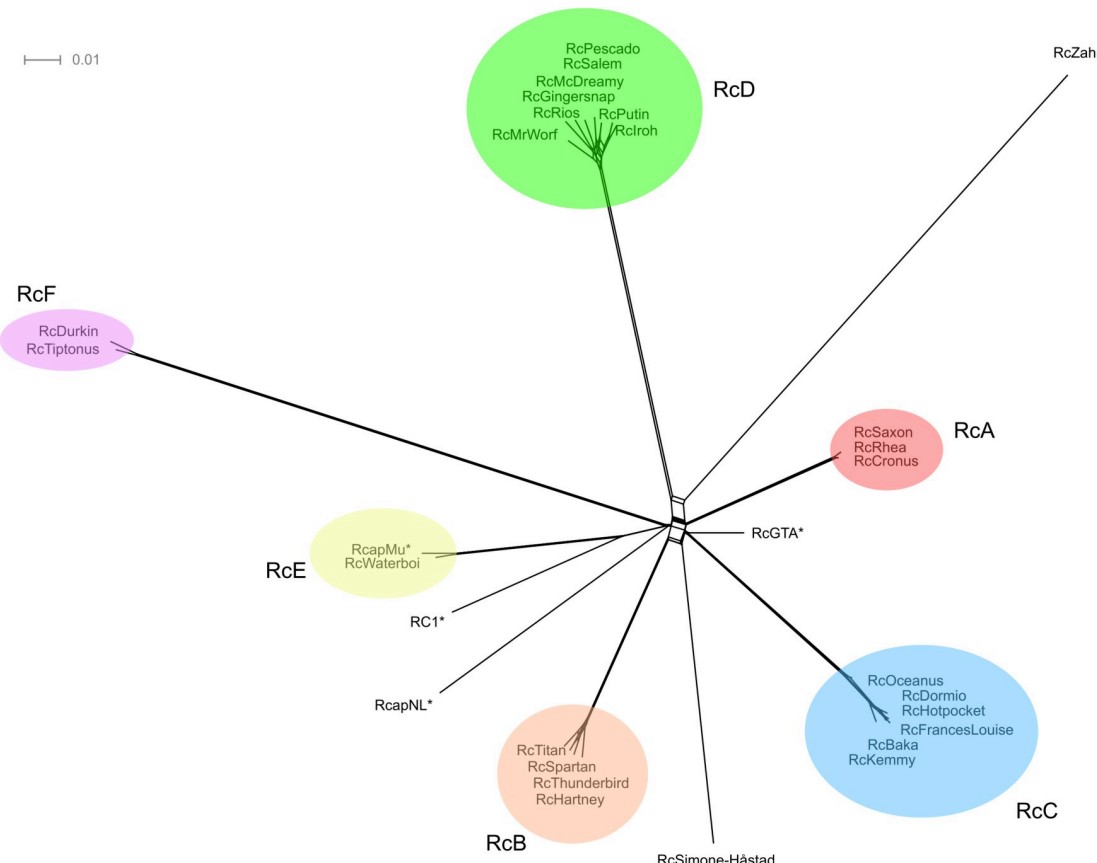

**Fig 2. Network phylogeny of *R. capsulatus* bacteriophages.** The predicted proteins of all 29 *R. capsulatus* phages and those found in the 14,087 bp RcGTA structural gene region were sorted into 833 families (phams) according to shared amino acid sequence similarities using Phamerator [31]. Each genome was then assigned values reflecting the presence or absence of members of each pham; the genomes were compared and displayed using Splitstree. Clusters are indicated with colored ovals. The scale bar indicates 0.01 substitutions/site. Asterisks indicate sequences available prior to this study.

**Table 5. Genes shared among *R. capsulatus* phages.**

| Cross-Cluster Shared Genes | Function |
|---|---|
| All RcC (RcOceanus 20), RcSimone-Håstad 5, RcZahn 33, RcGTA 18 | GTA TIM-barrel-like domain protein |
| All RcB (RcHartney 17), RcSimone-Håstad 77 | tail tube protein |
| All RcA (RcCronus 2), All RcD (RcPescado 24) | terminase large subunit |
| All RcB (RcHartney 7), RcSimone-Håstad 68 | terminase, large subunit |
| Both RcF (RcTiptonus 33), RcZahn 19 | major capsid protein |
| All RcB (RcHartney 16), RcSimone-Håstad 76 | minor tail protein |
| All RcB (RcHartney 20), RcSimone-Håstad 80 | minor tail protein |
| All RcA (RcCronus 14), All RcC (RcOceanus 17), RcSimone-Håstad 2, RcZahn 30, RcGTA 15 | minor tail protein |
| All RcA (RcCronus 16), All RcC (RcOceanus 18), RcSimone-Håstad 3, RcZahn 31, RcGTA 16 | minor tail protein |
| All RcA (RcCronus 17), All RcC (RcOceanus 19), RcSimone-Håstad 4, RcZahn 32, RcGTA 17 | peptidase |
| Both RcF (RcTiptonus 31), RcZahn 17 | capsid maturation protease |
| All RcA (RcCronus 3), All RcD (RcPescado 46) | endolysin |
| All RcA (RcCronus 42), RcapNL 67 | DNA primase |
| Both RcF (RcTiptonus 70), RcZahn 15 | DNA polymerase |
| All RcB (RcHartney 28), All RcD (RcPescado 8) | DNA polymerase |
| All RcE (RcWaterboi 19), Both RcF (RcTiptonus 14) | DNA binding, HU-like domain |
| All RcD (RcPescado 16), RcSimone-Håstad 15 | ribonucleotide reductase |
| All RcC (RcOceanus 55), RC1 25 | methylase |
| All RcD (RcPescado 53), RcZahn 95 | ThyX-like thymidylate synthase |
| All RcD except McDreamy (RcPescado 55), RcZahn 101 | ADP ribosyltransferase |
| All RcD (RcPescado 87), RcZahn 71 | AAA-ATPase |
| All RcC except RcOceanus (RcKemmy 35), RcSpartan 45, RcTitan 46 | NKF* |
| All RcA (RcCronus 27), RcMcDreamy 90 | NKF* |
| All RcB (RcHartney 24), All RcD (RcPescado 41) | NKF* |
| RcSalem 86, RcZahn 102 | NKF* |
| RcTitan 45, RcMrWorf 89, RcGingersnap 89, RcRios 89 | NKF* |
| All RcC (RcOceanus 45), RcMcDreamy 89 | NKF* |
| RcDurkin 96, RcZahn 110 | NKF* |
| All RcA (RcCronus 22), RcSimone-Håstad 12 | NKF* |
| All RcD (RcPescado 10), RcZahn 37 | NKF* |
| All RcD (RcPescado 11), RcZahn 38 | NKF* |
| All RcD (RcPescado 12), RcZahn 39 | NKF* |
| All RcD (RcPescado 13), RcZahn 40 | NKF* |
| All RcD (RcPescado 14), RcZahn 41 | NKF* |

*NKF = no known function.

same groupings were observed. Using the D0 distance formula, ten genus level clusters and two family-level clusters were predicted by the VICTOR method.

## RcA cluster

The three members of the RcA cluster, RcCronus, RcSaxon, and RcRhea, were the first phages isolated in our laboratory. They came from three separate water samples collected from a stream near the local water treatment plant in 2012 and 2013. Each phage genome has a GC

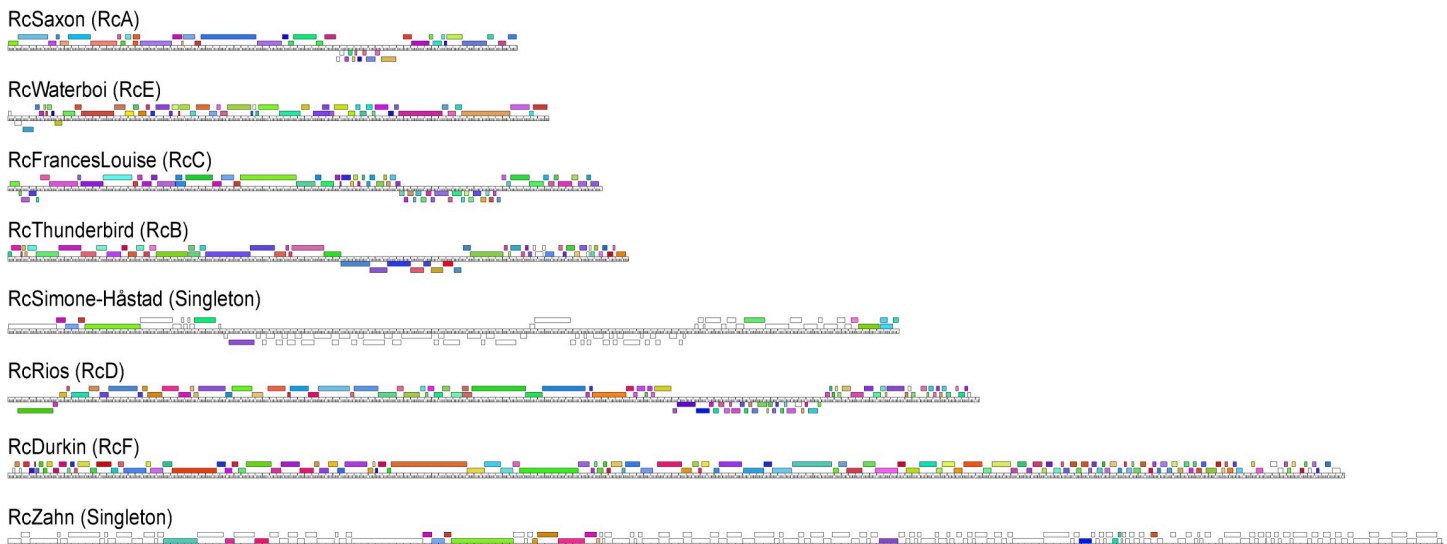

**Fig 3. Gene content phamerator maps of the longest genomes for each cluster and the newly discovered singletons RcSimone-Håstad and RcZahn.** Proposed genes transcribed from left to right are presented above the central graduated line while those transcribed in the opposite direction are found below the line. Boxes filled in with color represent genes that are shared by at least one other phage in the collection. Those without color represent orphams, or predicted genes that do not have any closely related homologues in this collection. The scale is the same for all genomes presented and they are ordered by length from shortest (RcSaxon, 36,081 bp) to longest (RcZahn, 101,599 bp).

content of 65.4% with either 45 (RcCronus and RcRhea) or 46 (RcSaxon) predicted genes. TEMs of these phages demonstrate the typical morphology associated with members of the *Siphoviridae* with an average capsid diameter of 59.7 nm and a tail length of 132.9 nm (Fig 1 and Table 1). The mean genome length for this cluster is 36,044 bp with 96 bp separating the largest (RcSaxon) and smallest (RcCronus) genomes. This average genome length is the smallest of any of the identified clusters while the average GC content is the highest. As expected with small genomes, these phages also have the smallest capsid diameters of any in this collection. The genomes of RcA phages have defined ends with 13 bp 5'overhang suggesting a cohesive end packaging strategy. Compared to the other clusters, their genomes have a relatively high nucleotide identity and except for one area align very closely (S2 Fig). RcRhea and RcCronus are nearly identical at the nucleotide level (99.23% ANI), while RcSaxon shows slightly larger sequence differences (99.15% ANI with RcCronus and 98.60% ANI with RcRhea).

The core set of genes for this cluster is comprised of 44 genes. RcRhea and RcCronus have the exact same set of 45 genes while RcSaxon has two novel genes (orphams) where the other two have just one (gene 25 in both RcRhea and RcCronus; genes 25 and 26 in RcSaxon). Genome sequences of these phages were reported in a genome announcement [30] and led to the creation of a phage genus, *Cronusvirus*, with RcCronus considered the type phage for the genus [28].

Organization of the genes in RcA phages match the typical organization seen in other tailed phages with the left side encoding structural proteins in a canonical order with the exception of endolysin (gene 3) placement. The right side contains a number of genes associated with DNA metabolism.

Members of the RcA cluster share 8 genes with other sequenced *R. capsulatus* phages outside of this cluster with 6 of these having known functions (Table 5). All members of this cluster share the large terminase subunit and endolysin with all of the RcD phages. Interestingly however, they also share a gene of unknown function with just one member of the RcD cluster, RcMcDreamy (gene 90). A set of three genes encoding two minor tail proteins and a peptidase in these phages is one of the most widely shared segments in this collection as it is found in all

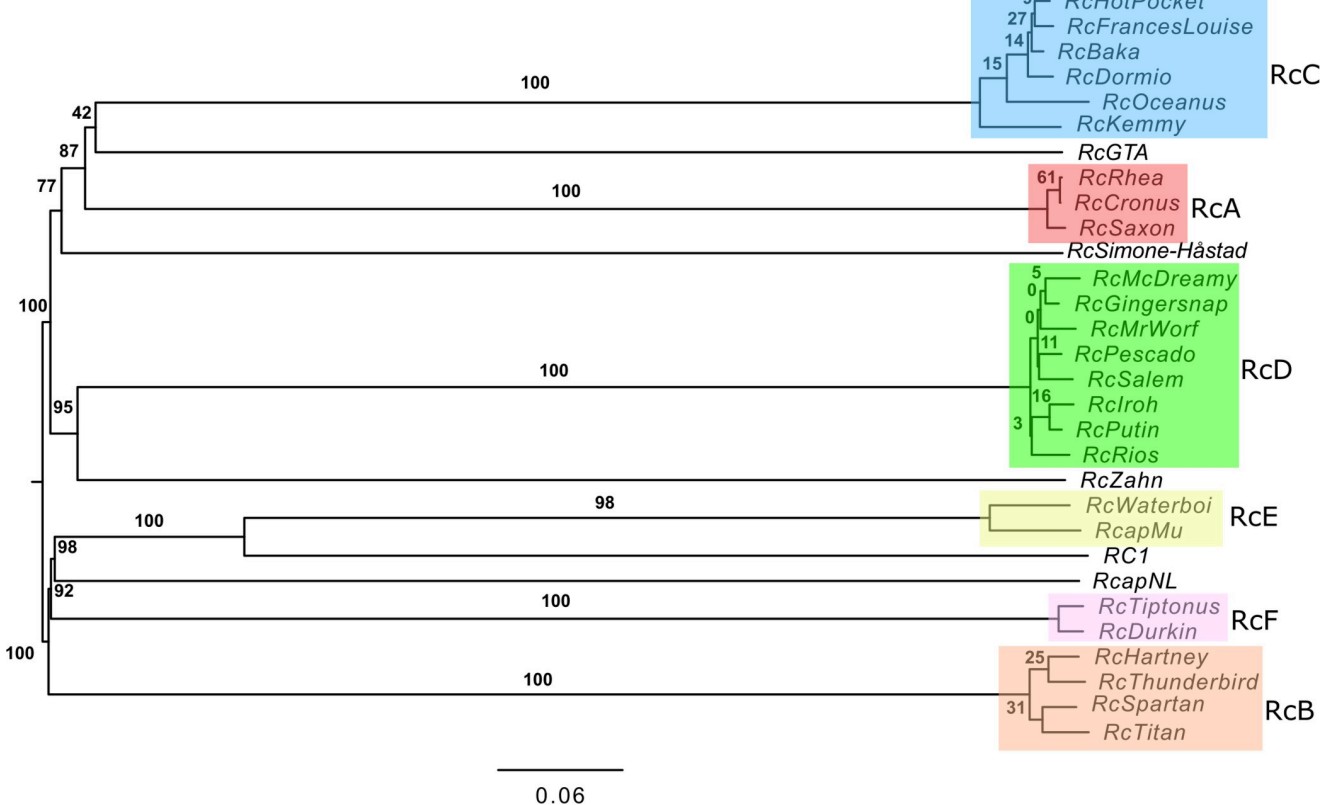

**Fig 4. Evolutionary relationships of *R. capsulatus* bacteriophages and RcGTA.** The genome BLAST distance phylogeny (GBDP) tree generated using phage nucleotide sequences entered into the VICTOR web application under settings recommended for prokaryotic viruses and the D0 distance formula. The numbers above branches are GBDP pseudo-bootstrap support values from 100 replications. The branch lengths on the resulting VICTOR tree are scaled in terms of the respective distance formula used.

RcA and RcC phages along with the singletons, RcSimone-Håstad and RcZahn, and RcGTA. The DNA primase of RcapNL (gene 67) is also shared with all members of the RcA cluster and is the only gene this phage shares with any others in this collection. Lastly, gene 12 of RcSimone-Håstad with no known function is shared by all members of this cluster (gene 22).

A recently published genome of a phage isolated off the coast of China that infects *D. shibae*, vB_DshS-R4C (GenBank accession MK882925.1), was reported to have substantial similarities to the genomes of the RcA cluster phages [33]. This same phage also was found to have the most similar major capsid and large terminase proteins through BLASTP searches with these sequences (Table 4). In this paper they propose that vB_DshS-R4C should be placed in the *Cronusvirus* genus. Fifteen of the 49 predicted genes of vB_DshS-R4C are shared with the RcA phages along with complete conservation of gene order. It was also noted that vB_DshS-R4C has an identified integrase whereas none of the members of the RcA cluster do. Despite the similarities of these phages to vB_DshS-R4C, RcA phages were found to be the most limited in their plaquing abilities and were only capable of forming plaques on the isolation strain, YW1, and none of the others examined including *D. shibae* (Table 2).

## RcB cluster

The four members of the RcB cluster, RcTitan, RcSpartan, RcThunderbird, and RcHartney, were isolated from a number of locations in Illinois and from Vancouver, Canada. RcTitan,

RcSpartan, and RcThunderbird were each isolated from independent water samples taken from a stream near a water treatment plant or water directly from a water treatment plant. RcTitan and RcSpartan samples were obtained in Bloomington, IL while RcThunderbird was from a wastewater treatment plant in Canada. RcHartney, however, came from an Illinois River location not immediately associated with a water treatment plant. The plaques formed by these phages on the isolation host YW1 are notably clear with well-defined borders and bio-informatic information is consistent with them being virulent phages. When examined on alternative hosts, they were found to produce similarly robust plaques on the St. Louis and B10 strains with somewhat cloudier plaques on YW2 and B6. They were unable to form plaques on 37B4, Iona, *R. pomeroyi*, or *D. shibae* (Table 2).

A representative transmission electron micrograph of these phages demonstrates they are noncontractile tailed phages with an average tail length of 150.2 nm and icosahedral heads with an average diameter of 61.0 nm (Fig 1 and Table 1). Two of these phage genomes were described in a genome announcement and led to creation of the genus *Titanvirus* with RcTitan serving as the type phage [29].

RcB phage genomes are circularly permuted and use a headful packaging mechanism. The mean GC content is 54.9% and the mean genome length is 44,040 bp with a 986 bp difference between the largest (RcTitan) and smallest (RcHartney) of these genomes. RcHartney has 60 predicted genes while each of the other three have 61. All members of this cluster share a core set of 51 genes with 10 out of the total of 72 genes in this cluster being orphams. These genomes share strong sequence identity through the first 32 kb with identities ranging from 93% to 97% in this region and identical gene content and order (S3 Fig). Two different genes are present as gene 36. Interestingly, one version of gene 36 is shared by RcSpartan and RcHartney and has been annotated as a helix-turn-helix DNA binding domain while the other longer version of gene 36 is shared between RcTitan and RcThunderbird and is annotated as an endonuclease. As RcTitan and RcSpartan were isolated from the same geographical location this suggests that the version of gene 36 is not correlated with the geographical location of the water sample. Further to the right in these genomes there is greater variation in the genes present and the sequence.

Members of RcB share genes with RcSimone-Håstad, RcC, and RcD phages, with conservation primarily at the amino acid level, not at the nucleotide level (Table 5). RcB phage genes 7, 16, 17, and 20 (encoding the terminase large subunit, minor tail protein, tail tube protein, and minor tail protein respectively) are shared with RcSimone-Håstad (genes 68, 76, 77, and 80). Another gene with no known function is found in only two members of the RcB cluster, RcTitan 45 and RcSpartan 46, and almost all of the RcC phages (though not RcOceanus); represented by RcKemmy gene 35. RcB phage genes 24 and 28 (No known function and DNA polymerase respectively) are shared with all of the RcD phages (RcPescado genes 41 and 8). Lastly, a gene with no known function in a single member of this cluster, RcTitan gene 45, is shared by just three of the eight members of the RcD cluster, RcMrWorf, RcGingersnap, and RcRios,—gene 89.

Outside of the *R. capsulatus* phages, the major capsid and large terminase BLASTP analysis revealed RcB cluster phages share some similarity with *Pseudomonas aeruginosa* phage vB_PaeS_C1 (accession number MG897800) and the *E. coli* phage Halfdan (accession number MH362766) (Table 4). Both isolated from wastewater environments.

## RcC cluster

Members of the RcC cluster have all been isolated from water samples in Illinois. The 6 members, RcOceanus, RcDormio, RcBaka, RcFrancesLouise, RcHotpocket, and RcKemmy, were

isolated over a span of four years, with RcOceanus isolated first in 2013 and RcKemmy isolated most recently, in 2018. Plaques formed by members of this cluster are generally turbid. All members except RcOceanus are able to form plaques on YW1 (the isolation host) and B10 with barely discernable plaques on St. Louis and no plaques seen with the other potential hosts tested. RcOceanus only forms plaques on YW1 (Table 2).

A representative transmission electron micrograph of these phages reveals they are *Siphoviridae* with average tail lengths of 115.8 nm and icosahedral heads with an average diameter of 66.2 nm (Fig 1 and Table 1).

The genomes of RcC phages have defined ends with 11 base pair 5'overhangs suggesting a cohesive end packaging strategy. Mean GC content is 64.1% while mean gene content is 67 and mean genome length is 41,023 bp. All members share a core set of 53 genes with just 4 of the 76 genes in this cluster being orphams. The mean gene number and genome length are skewed by the RcOceanus genome being much smaller than the rest (Table 3). RcOceanus has twelve fewer genes and is 3,736 bp smaller than the next smallest genome (RcKemmy) as seen in the alignment of RcC cluster phages (S4 Fig). This is of particular interest given the difference in host range observed for RcOceanus.

The organization of genes is typical of most tailed phages with those for structural proteins to the left end. Members of this cluster appear to have a fused major capsid and protease. There is also evidence for a holin/endolysin cassette just to the right of the tail protein genes. A small open reading frame between the holin and endolysin could be either another holin or an antiholin but was left unlabeled in the absence of experimental evidence. The right ends of these genomes contain a number of genes encoding proteins with predicted DNA binding domains as well as proteins that are involved in nucleotide metabolism such as a ribonucleotide reductase and a nuclease.

As noted above, the RcC phages share a set of adjacent genes associated with tail components with the RcA phages, two singletons and RcGTA as well as the conservation of a gene between two members of the RcB cluster and most of the RcC phages (gene 35 in RcKemmy). Another interesting relationship is a gene with an unknown function present in all RcC phages (RcOceanus gene 45) that is present in a single representative of the RcD cluster, RcMcDreamy (gene 89). Additionally, two genes that are conserved in all RcC phages (RcOceanus 55 and 20) are shared with RC1 (gene 25, a DNA methylase) and RcSimone-Håstad 5, RcZahn 33, and RcGTA 17 encoding a GTA TIM barrel-like domain tail protein.

BLASTP analysis of the major capsid and large terminase protein sequences found the best matches to these proteins are in the *Stenotrophomonas maltophilia* phage S1 (GenBank accession number NC_011589) and the *Wolbachia* phage WOVitA1 (GenBank accession number HQ906662) respectively.

## RcD cluster

The RcD cluster has the greatest number of isolated phages in our sequenced collection. The eight phages were collected starting in 2016. These phages produce cloudy plaques and apparent lysogens can be isolated that are resistant to superinfection and produce infectious particles when the cells are grown to stationary phase. Members of this cluster have a relatively broad ability to form plaques on YW1, YW2, B6, B10, and St. Louis strains though the plaques formed on the latter four hosts tend to be much cloudier than those on YW1. None of them can form plaques on 37B4, Iona, or the marine hosts *D. shibae* and *R. pomeroyi* (Table 2).

The transmission electron micrograph of cluster RcD phage RcMcDreamy is consistent with the others obtained in this cluster and demonstrates they are somewhat larger noncontractile tailed phages with an average tail length of 204.4 nm and icosahedral heads with an

average diameter of 73.3 nm (Fig 1 and Table 1). The larger head size observed is consistent with the larger genome size for these phages.

Genomes in the RcD cluster have a mean length of 68,058 bp, a mean GC content of 60.2%, and a mean gene content of 101 predicted genes (Table 1). 1,280 bp separate the smallest genome in this cluster RcPescado (with 99 predicted genes) from the largest, RcRios (with 103 predicted genes). The core genome for this group consists of 81 genes with 14 out of 125 total genes being orphams. The genome ends have a 5' overhang of 12 bases. There is a significant amount of repeat sequence at the right end of the genomes that also correlates with sequence and gene diversity (S5 Fig). The left ends of the genomes are highly conserved but unlike most of the other sequenced *R. capsulatus* phages there are several genes involved in replication (DNA primase, DNA polymerase, DNA helicase), nucleotide metabolism (ribonucleotide reductase, nuclease) and lysogeny (tyrosine integrase, excise, immunity repressor) on this end of the genomes. These genes are most commonly on the right ends of the genomes. The structural genes at the left end start at gene 23 in all members of the cluster, beginning with the small terminase subunit and are in a very typical order, just shifted further into the genome.

Most shared genes present in the RcD cluster have already been described above with the other clusters with the exception of genes shared with singletons RcSimone-Håstad and RcZahn (Table 2). The ribonucleotide reductase found in RcSimone-Håstad (gene 15) is also found in all of the RcD phages (gene 16). RcZahn shares 9 genes with at least one member of the RcD cluster. RcZahn genes 37–41, 71, and 95 are present in all of the RcD phages. Predicted functions are only available for genes 71 and 95; AAA ATPase and ThyX respectively. The RcD homologs of RcZahn 37–41 are in the same order in RcZahn and are in the highly conserved region on the left side of the genome (RcD genes 10–14). Another gene encoding an ADP ribosyltransferase is present in RcZahn (gene 101) and in 6 of the 8 RcD phages but is absent from RcMcDreamy and RcSalem. The final shared gene is RcSalem gene 86 and RcZahn gene 102 which is not found in any other sequenced *R. capsulatus* phage and has no known function.

The NCBI BLASTP best hits for the major capsid and large terminase protein sequences for this cluster were with the *Ruegeria pomeroyi* phage vB_RpoS-V18 (GenBank accession number NC_052970) and the *Loktanella* sp. CB2051 phage pCB2051-A (GenBank accession number NC_020853) (Table 4). These isolation hosts are both marine Rhodobacteraceae and these phages were both isolated from brackish or marine environmental samples.

## RcE cluster

There are two members of the RcE cluster, RcapMu and RcWaterboi. RcapMu is integrated into the genome of B10 and was isolated by heat treatment to encourage excision from the genome by Fogg et al. [15], though it has likely been present in the genome of the original B10 strain since isolation in 1974 [15, 34]. We also isolated RcapMu from B10 by growing it to late stationary phase and using filtrate to infect YW1. RcWaterboi was isolated from a water sample in 2016 indicating that this type of phage continues to circulate.

As the name RcapMu suggests, these phages share characteristics with *E. coli* phage Mu. Mu type phages are often present as prophages in the host bacterium but when induced to excise, they use transposition throughout the genome as the mechanism for replication. Both of the RcE cluster phages easily form lysogens and when cross-infection experiments were attempted, the RcapMu lysogen was immune to RcWaterboi, and the RcWaterboi lysogen was immune to RcapMu. Both phages formed plaques on YW1, YW2, and St. Louis strains of *R. capsulatus*. Neither was able to form plaques on B6, B10, 37B4, or Iona (Table 2). The inability to infect B10 was expected, since it contains an integrated RcapMu prophage. Neither of these phages could form plaques on *D. shibae* or *R. pomeroyi*.

Sequenced DNA from RcWaterboi revealed the ends of the genome were heterogeneous as expected for a phage that replicates by transposition. The sequence reported includes only the sequence where the heterogeneity ceased. It aligns well with RcapMu as the two phages share 90% ANI. The GC% for both is 64.8. The mean length is 38,792 bp, and the mean gene number is 57. The two phages share 51 genes with 7 being unique to RcapMu and 4 being unique to RcWaterboi.

The organization of the structural genes follow the order commonly observed in members of the *Siphoviridae* but with terminase genes close to the center of the genome (gene 30 in RcWaterboi and gene 32 in RcapMu; S6 Fig). Genes to the left in these genomes are involved in the transposition process, regulation of lysogeny, and lysis. Another *R. capsulatus* phage, RC1, isolated in 2002 and sequenced by the Broad Institute, shares 12 of its 56 genes with the RcE phages and is included in S6 Fig to emphasize the shared regions. Though the ANI is only 59% between RC1 and either of the two RcE phages, there is significant amino acid sequence similarity with some of the putative tail proteins and several other genes that are homologs. Additionally, the order and placement of these genes along the genome is largely conserved. Overall enough difference exists that RC1 is not considered a member of this cluster however. One of the genes shared with RC1 is the only other gene in these phages that is shared with members outside this cluster. This gene is shared with the RcF cluster (Table 5; RcWaterboi gene 19) and has an HU-like domain suggesting it is involved with DNA organization.

While the *E. coli* Mu is a *Myoviridae* with a contractile tail, transmission electron micrographs of RcapMu had previously shown that this phage has a flexible, noncontractile tail [15]. In our experiments the tails of RcWaterboi were easily visualized (Fig 1) as flexible, noncontractile structures placing these phages in the *Siphoviridae*.

BLASTP searches of the major capsid and large terminase protein sequences revealed the most similar major capsid and large terminase proteins were both found in *Rhizobium radiobacter* phage RR1-B (GenBank accession number NC_021557). RR1-B is described as a *Myoviridae* isolated from a sample of an upwelling of deep-biosphere sediment in the open equatorial Pacific.

## RcF cluster

The two members of this cluster, RcDurkin and RcTiptonus, were isolated from water samples taken in Illinois. They produce notably small plaques on YW1, and are able to also form plaques on YW2, B6, B10 and St. Louis strains of *R. capsulatus*, but not on the 37B4 or Iona strains or on *D. shibae* or *R. pomeroyi*. These phages have a typical structure with an icosahedral capsid diameter average of 80.6 nm and flexible tail with an average length of 297.2 nm (Fig 1 and Table 1). There is no bioinformatic or laboratory evidence for lysogeny, so the RcF phages appear to be virulent.

When genome sequencing of these phages was completed the nature of the ends were initially difficult to determine. Phageterm [35] on the Galaxy cluster at the Pasteur Institute (https://galaxy.pasteur.fr) deduced that these phages use a packaging strategy similar to P1. This involves a defined start site with cleavage at a *pac* site and then headful packaging with the downstream *pac* site modified by methylation to prevent cleavage. The result is that the left end is well defined, but the right end is distributed over a wide range, but duplicates sequence found at the left end. The genome sequences reported include the whole genome just once. GC content for both phages is 57.8% and the mean gene content is 140 with a common core set of 133 genes with RcDurkin having 6 orphams and RcTiptonus having 5. The mean genome length is 94,635 bp, making this the cluster with the longest genomes (Table 3). 548 bp separate the larger from the smaller genome.

The general organization of these genomes is similar to most members of the *Siphoviridae* with the structural proteins on the left end and DNA replication and metabolic genes to the right. The terminase is gene 25, and the canonical organization of structural genes follows through the tape measure followed by presumptive tail proteins (S7 Fig). As mentioned previously, gene 14 in these phages is a predicted HU DNA binding protein shared with RcE and RC1 phages (Table 5). The only other genes shared with a phage outside of the cluster are shared with the singleton RcZahn. The arrangement of genes 31 and 33 in the RcF phages is similar to genes 17 and 19 in RcZahn and encode the capsid maturation protease and major capsid respectively. Gene 70 in the RcF phages is similar to gene 15 in RcZahn and predicted to encode a DNA polymerase. Lastly, RcDurkin gene 96 (no known function) shares homology with RcZahn gene 110 but this gene is not present in RcTiptonus.

## Singletons

**RcSimone-Håstad.**   This phage was isolated from a water sample collected from a stream in the Swedish village of Håstad using the host strain SB1003 (a derivative of B10). Its structure is different from all of the others reported here, as it has a slightly prolate head. Average tail length is 184.2 nm with a head width of 54.1 nm and length of 76.7 nm (Fig 1 and Table 1). Along with infecting its isolation host strain and B10 it was also found to infect YW1 and form plaques on St. Louis, but was unable to form plaques with YW2, B6, 37B4, Iona, *D. shibae*, and *R. pomeroyi*.

Sequencing of RcSimone-Håstad revealed a genome of 63,102 bp and 60.75% GC with a 77 bp terminal repeat (Table 1). It is predicted to have 80 genes with genome ends in the middle of the structural gene region; the left end starts with the tape measure gene and is followed by several tail protein genes while the right end contains the terminase to tail assembly chaperone region of the structural genes (S8 Fig). As noted above, RcSimone-Håstad shares genes with several phages in other clusters, with the greatest number of genes shared with the RcB cluster (Table 5). There is no evidence for any genes associated with lysogeny and no lysogens were isolated when attempted so this phage appears to be virulent.

BLASTP analysis of the RcSimone-Håstad major capsid and large terminase subunit proteins yielded best hit matches with two *Pseudomonas aeruginosa* phages, YuA (GenBank accession number AM749441) and PaMx28 (GenBank accession number JQ067089) (Table 4). Both of these share the somewhat unique slightly prolate capsid morphology seen with RcSimone-Håstad [36, 37].

**RcZahn.**   RcZahn was isolated from a water sample collected in Illinois, USA in 2018. This phage has a typical structure with a tail length of 296.9 nm and a head diameter of 87.1 nm (Fig 1 and Table 1). Notably this capsid diameter is the largest of any of the phages described here and the tail length is just below those of the RcF phages. While its ability to form plaques on the examined *R. capsulatus* strains is limited to just YW1, B6, and 37B4 it is the only phage in this collection capable of forming plaques on 37B4 or *D. shibae*. Lysis on these two hosts only occurred with aliquots of high titer (greater than $10^8$ PFU/ml) lysates placed directly onto cells embedded in top agar. Further cultivation of this phage on either of these strains was not possible (Table 2). Despite the fact that it can lyse cells of *D. shibae*, it was unable to form plaques on the other marine host examined, *R. pomeroyi*. No genomic or phenotypic evidence exists for the formation of lysogens by this phage so it appears to be virulent.

The genome of RcZahn is the largest of any *R. capsulatus* phages reported here at 101,599 bp. The % GC is 60.7 (Table 3) and it is predicted to contain 147 genes, all of which would be transcribed in the same direction (S9 Fig). The sequence data suggest a circularly permuted genome with headful packaging. The general order of structural genes placed on the left is

consistent with most tailed phages, but is interrupted with numerous genes associated with DNA metabolism and genes without a predicted function between the terminase and portal genes (S9 Fig). Unlike most of the phages described here, there is no clear evidence for sequences associated with the programmed translational frameshift usually found in the tail assembly chaperone genes. As noted above, RcZahn shares some genes with several other phages in this collection, with the greatest number shared with the RcD cluster (Table 5).

BLASTP searches with the RcZahn major capsid and large terminase subunit protein sequences yielded best hits with the *Rhizobium* sp. R693 phage RHph_TM16 (GenBank accession number MN988459) and *Stenotrophomonas maltophilia* phage vB_SmaS_DLP_3 (GenBank accession number MT110073) (Table 4). These two phages have similarly sized genomes as RcZahn and originated from soil samples.

*Phage collection's relationship to RcGTA*. As noted in several instances above, the RcGTA structural gene region shares a set of three to four genes with high amino acid sequence similarity to those of many of the phages in this collection (RcA and RcC phages and the singletons, RcSimone-Håstad and RcZahn, Fig 5). Nucleotide sequence conservation in these regions is quite low. These genes are annotated as the minor and major tail proteins, the cell wall hydrolase, and a "GTA TIM-barrel-like domain" which is associated with RcGTA tail proteins [18]. A recent paper has renamed the minor tail protein as the distal tail protein, the second tail protein as the hub protein, the cell wall hydrolase as a peptidase and the GTA TIM-barrel like domain protein as megatron [18]. It is also notable that none of the other genes found in this region (RcGTA 1–13) have been found to have close homologs with any of the other genes in this collection.

In addition to the 14,087 bp region encoding RcGTA structural proteins, several other genes elsewhere in the *R. capsulatus* SB1003 genome have been shown to be involved with RcGTA production [38]. These four regions designated by their locus identifiers as rcc00171 (encoding tail fibers), rcc00555 and rcc00556 (endolysin and holin respectively), rcc01079 and rcc01080 (GhsA and GhsB head spikes/fibers), and rcc01865 (GafA transcriptional regulator of RcGTA) and rcc01866 (unclear function, possibly capsid maturation) were also used as queries in this database. Of these only rcc01080, meets the 32.5% CLUSTALW threshold used by phamerator for pham memebership with genes found in these newly isolated phages [18, 39]. This gene is found in the RcE phages RcWaterboi and RcapMu. This match with RcapMu as well as a match between rcc01079 and RcapNL have previously been reported [39].

## Discussion

This work represents the first major survey of environmentally-isolated phages of *R. capsulatus* in more than 45 years. The lack of more recent reports may relate to the challenges of using morphology and host range as the primary means of classifying phage relationships as used in the report from 1975 describing 95 phages and 16 potential clusters [14]. The advent of relatively inexpensive sequencing of phage genomes has dramatically altered the ability to look at phage relationships. The sequencing of phage genomes combined with the integration of phage discovery into the biology courses of authors RMA and DWB led to development of the collection of sequenced *R. capsulatus* phages described here.

With this collection of 26 newly isolated phages we have used genomic sequence and protein conservation to identify six distinct clusters and four additional singletons that likely represent a small proportion of the overall diversity of phages that infect this host. Studies with gene transfer agents have demonstrated that lateral gene transfer events are possible via theses agents and that these events may have effects on bacterial adaptation and evolution [40, 41]. Since phages can also transfer host DNA, understanding the genomic diversity of *R. capsulatus*

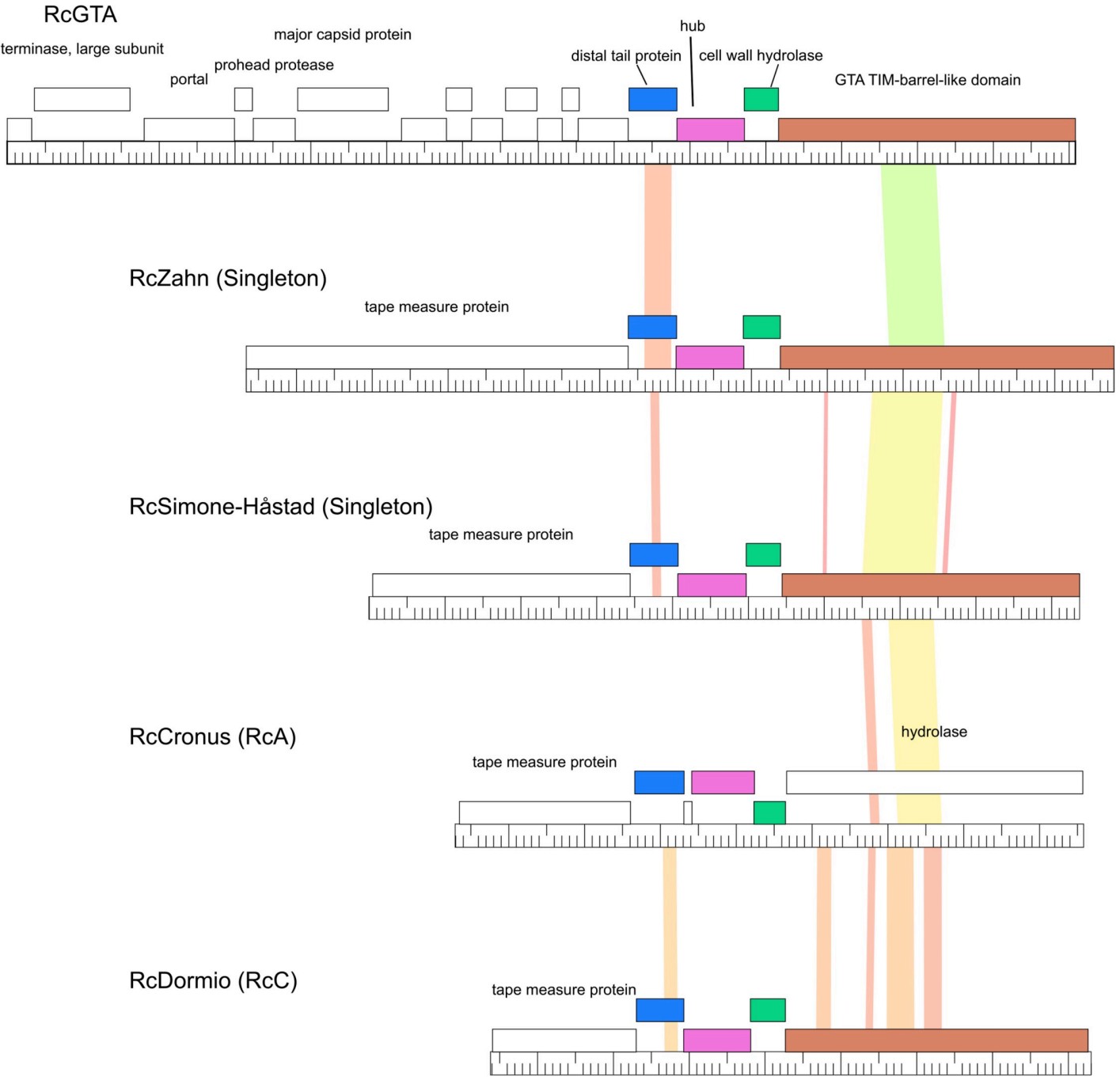

**Fig 5. Genes shared with RcGTA.** Genes shared between RcGTA, RcZahn, RcSimone-Håstad, and members of the RcA and RcC clusters are indicated by shared color. The locations of several of these proteins within the tail region of RcGTA particles have recently been identified [18]. Genes sharing the same coloration have been designated by Phamerator as being members of the same pham while those that are colorless are considered to be orphams as their amino acid sequence identity with any other predicted proteins in this database does not reach the 32.5% identity CLUSTALW threshold used by Phamerator. Regions of color between genomes highlight pairwise nucleotide sequence similarities. Colors towards the red end of the spectrum represent regions with reduced levels of similarity but still above the BLASTN cut off value of $10^{-4}$ that still warrants shading.

phages could provide insight into the readily accessible genetic material for incorporation into the bacterium.

Previous studies have shown that several RcGTA genes are found in the genome of other bacterial species in the α-proteobacteria including the Caulobacterales, Rhizobiales, Rickettsiales, Sphingomonodales, and Rhodospirallales [42].Within the Rhodobacterales, many GTA containing bacteria are in the marine roseobacteria including members of the Ruegeria genus. RcGTA-like genes have also been identified in phages ΦJL001 and RDJLΦ1 that infect Rhizobiales [43] and *Roseobacter denitrificans* [44].

We constructed a database that includes the RcGTA structural gene cluster, the genomes of previously sequenced *R. capsulatus* phages, and the 26 new phage isolates that has 1,563,838 bp of DNA sequence encoding 2,350 predicted genes that can be grouped into 833 phams with 367 orphams. While all of these isolates possess a *Siphoviridae* morphology, each grouping has distinct characteristics that clearly delineates it from the others. These features include particle morphology, plaque morphology and host range, but most importantly genomic characteristics. All of the clusters and singletons share genes with at least one other phage type and most commonly with several other types–a pattern commonly seen with studies of phages [45]. Additionally, some of the most widely shared genes in this collection are shared with RcGTA, consistent with it having originated from a phage. It is notable however, that the majority of genes known to be involved with RcGTA do not appear to be closely related to homologs in this collection. The genes that are shared between RcGTA and some of the newly isolated phages are clearly linked to host infection: distal tail protein (gene 14), hub (gene 15), peptidase (gene 16), and megatron (gene 17) are proteins that are involved with the interface between phage and host with the distal tail protein proposed to be involved in recognition, the hub protein having domains for carbohydrate binding, the peptidase protein degrading peptidoglycan, and megatron having a domain involved in penetration. Since RcGTA delivers phage DNA to the periplasmic space [46] this may suggest a similar infection route for phages with the megatron gene (formerly described as a GTA TIM barrel domain protein).

The lack of shared genes between RcGTA and the phage collection could simply be due to the limited nature of this collection, or it could be due to the fact that the host used for the majority of the isolations for this study, *R. capsulatus* strain YW1, cannot be transduced by RcGTA and thus related phages that utilize similar mechanisms of infection may be selected against.

The metabolic flexibility of *R. capsulatus* has been studied extensively and led to an increased understanding of photosynthesis and gene regulation in response to environmental factors such as light and oxygen presence. Characterization of RcGTA led to the use of this agent for genetic manipulation of the host. The work presented here is an initial description of the genomic diversity of phages that infect *R. capsulatus*. In addition, it provides suggestive evidence that phages that infect members of the Rhodobacteraceae may move between different host species, including between marine and freshwater environments. It does not provide evidence for any photosynthesis related genes carried in the genome of these phages. Continuing to investigate the breadth of diversity within the phages of *R. capsulatus* will illuminate the reservoir of genes available to members of this family of bacteria.

## Materials and methods

### Growth and isolation

The majority (15 out of 26) of the newly isolated phages described here were isolated by students in the Illinois Wesleyan University Science Education Alliance-Phage Hunters Advancing Genomics and Evolutionary Science (SEA-PHAGES) course using the curriculum and protocols developed for this program and adapting them for use with *R. capsulatus* (S2 File) [47]. Phages were isolated by direct plating or enrichment on the *R. capsulatus* hosts YW1 C6,

a tetracycline-resistant derivative of strain YW1, and SB1003 [34, 48]. Cells were grown for 2–3 days under aerobic conditions at 30˚C in YP (0.3% yeast extract and 0.3% peptone), YPS (YP supplemented with 2 mM $CaCl_2$ and 2 mM $MgSO_4$), or PYCa (0.1% yeast extract, 1.5% peptone, 0.5% $CaCl_2$, 0.1% glucose) to an $OD_{600}$ of ~1.0. Phages were plated on solidified versions of these media in a 0.4% top agar overlay and purified by three rounds of single-plaque replating after initial plaque identification. Plaques most typically appear after two days of incubation at 30˚C but could sometimes be visible after one. No additional plaques appear after two days of incubation. Phages were then amplified by confluent plating on bacterial lawns for increasing titer which was then used for TEM, DNA isolation, and additional experimentation such as lysogen testing and host range determination.

## Lysogen testing

The ability of phages to enter into a lysogenic cycle was examined through use of a large spot plate assay using *R. capsulatus* strain YW1. Briefly, a 100 μl aliquot of a high titer lysate ($>10^8$ pfu/ml) was placed onto the surface of a YP, YPS, or PYCa petri dish overlaid with solidified 0.4% top agar with *R. capsulatus* cells. This plate was then incubated for one week at 30˚C. A sample of the resulting plaque was then taken using an inoculating loop and was streaked for isolation onto a fresh plate. After 3–7 days of incubation, when bacterial colonies had grown to a reasonable size for further cultivation, the resulting colonies were again streaked for isolation. From these plates individual colonies were chosen to start liquid cultures that were incubated for 3–7 days for use as the inoculum for a spot plate assay (at an $OD_{600}$ between 0.8 and 1.0). Most typically, this was three days but could sometimes be longer for slower growing cultures. These cultures were then challenged in a spot plate assay with 10ul of the original phage lysate to check its susceptibility to infection. For any isolates that did not support plaque formation, a 1ml sample was centrifuged in a microcentrifuge to pellet the cells and the resulting supernatant was then assayed for the presence of phage on plates containing solidified top agar containing *R. capsulatus* cells naïve to the challenging phage. Isolates that were unable to support cultivation of their challenging phage and exhibited phage release were considered presumptive positives for lysogens and the phages they harbored were considered to be temperate. Potential lysogeny was also assessed bioinformatically by looking for the presence of genes encoding integrase and excise proteins.

## Host range determination

To examine the ability of newly isolated phages to form plaques on alternative hosts seven different strains of *R. capsulatus*, YW1 [34], YW2 [34], B6 [34], B10 [34], St. Louis [34], 37B4 [49], and Iona (an isolate of the Beatty lab), and two marine Rhodobacteraceae, *D. shibae* DFL12 and *R. pomeroyi* DSS3 were used as potential hosts in spot assays with high titer ($>10^8$ pfu/ml) lysates. Briefly, 10 μl aliquots of these high titer lysates were spotted onto plates with cells of these strains embedded in solidified top agar (0.4% agar in YPS). Plates were incubated for three days at 30˚C and were then scored for the formation of plaques.

## Sequencing and annotation

DNA was extracted from concentrated phage samples using a modified protocol with the Wizard DNA Clean-Up Kit (Product #A7280; Promega, Madison, WI). Briefly, RNaseA and DNase I was added to 1.4 ml of high-titer ($>10^8$ pfu/ml) plate-lysate-collected phage sample and incubated at 37˚C for 10 min. This mixture was then added to 2 ml of DNA clean-up resin, mixed well, and then split between two clean-up columns. Three 2 ml washes of 80% isopropanol were applied to these columns which were then centrifuged to remove any remaining

wash solution. Genomic DNA was then eluted from the columns with two applications of 50 μl of water at 90˚C and centrifugation at 10,000 x g for 1 minute.

Sequencing was performed by ATGC inc., Wheeling, IL, The Sequencing Center, Fort Collins, CO, North Carolina State University, and the University of Pittsburgh. Sequences were determined by Illumina paired end reads and assembled using Newbler (Version 2.9) and Consed (Version 29) using the default settings. Fold coverage ranged from 86 to 6667 (Table 3).

Genome ends were predicted as described by Russell [50] by viewing the assembled sequence in Consed, using PAUSE and looking for features associated with ends. Phages with defined ends are readily identified by observing that all sequencing reactions terminated at a specific base on each end. Overhangs are also readily detected in this manner. The absence of these features along with sequencing reactions that connect the ends were used to determine if the genomes were circularly permuted. In the case of P1 type packaging observed in the RcF cluster, the genome ends were unclear using standard approaches so the genome was submitted to Phageterm using the default settings [35] on the Galaxy (v1.0.12) cluster at the Pasteur Institute (https://galaxy.pasteur.fr). Genomes were annotated using DNAMaster v.5.23.6 (cobamide2.pitt.edu) and PECAAN v.20210526 (discover.kbrinsgd.org) with analysis by Glimmer v.3.02, Genemark v.3.25, NCBI BLAST v.2.9.0, tRNAscan-SE v.2.0, Aragorn v.1.2.38, and HHpred v.3.2.0 informing decisions about gene location and function prediction during annotation. Default parameters were used for all software.

## Genome comparisons

PhamDB [51] was used to create a database with 30 entries (29 phage genomes and the ~14 kb RcGTA segment). These genomes were then analyzed using the Phamerator software package which uses an alignment-free algorithm, kClust, to group predicted gene products into "phamilies" based on related amino acid sequence and allows for comparison of genomes in terms of in-common gene presence and organization [31, 52]. The resulting database, Rhodobacter_capulatus at https://phamerator.org/, was used to identify similar predicted genes in the genomes and create images of aligned genomes. Formatting of these images was performed in Inkscape.

Gene content networks were created using Splitstree [26] based on pham membership of genes in each genome as determined by Phamerator with a 32.5% CLUSTALW identity cutoff. Analysis of nucleotide conservation to determine proposed evolutionary relationships was performed at the VICTOR site (https://ggdc.dsmz.de/victor.php,) [32].

## Electron microscopy

To negative stain the samples, 10 microliters of a high titer lysate sample were placed on carbon and Formvar coated 300 mesh copper grids. After 5 minutes the sample was wicked away with filter paper, so as to not disturb the attached sample, and replaced with 10 microliters of an aqueous solution of 2% uranyl acetate. The uranyl acetate was also wicked away so as to leave a thin film of the solution which was allowed to dry.

The dried grids were viewed with a JEOL company JEM 1010 TEM at 80 kV. Images were acquired using a Gatan MegaScan 794 digital camera.

Capsid diameter and tail length measurements were made using ImageJ [53] and reported values are the averages of measurements taken on at least three separate images (with three different phage particles) and multiple cluster members when possible.

## Supporting information

**S1 Fig. Dotplot comparison of catenated genomes.** A dotplot comparison of the catenated genomes against themselves was created using Gepard [1]. Areas of clustering are color-coded to match the cluster colors on Fig 2.
(TIF)

**S2 Fig. Genome organizations of *R. capsulatus* RcA cluster phages.** Genome maps of the RcA phages are shown. Pairwise nucleotide sequence similarities are displayed with spectrum-coloring between genomes, with violet representing greatest similarity and red the least similar, above a threshold E value of $10^{-3}$. Genes are represented as boxes above or below the genomes reflecting rightwards- and leftwards-transcription respectively. Genes are colored according to their phamily designations using Phamerator [31] and database Rhodobacter_-capsulatus.
(TIF)

**S3 Fig. Genome organizations of *R. capsulatus* RcB cluster phages.** See S2 Fig for details.
(TIF)

**S4 Fig. Genome organizations of *R. capsulatus* RcC cluster phages.** See S2 Fig for details. Areas where red lines appear between genome maps indicate the presence of repeat sequences.
(TIF)

**S5 Fig. Genome organizations of *R. capsulatus* RcD cluster phages.** See S2 Fig for details.
(TIF)

**S6 Fig. Genome organizations of *R. capsulatus* RcE cluster phages and RC1 phage.** Genome maps of the RcE phages along with the singleton RC1 are shown. See S2 Fig for details.
(TIF)

**S7 Fig. Genome organizations of *R. capsulatus* RcF cluster phages.** See S2 Fig for details.
(TIF)

**S8 Fig. Genome organization of *R. capsulatus* phage RcSimone-Håstad.** See S2 Fig for details.
(TIF)

**S9 Fig. Genome organization of *R. capsulatus* phage RcZahn.** See S2 Fig for details.
(TIF)

**S1 File.**
(DOCX)

**S2 File.**
(DOCX)

## Acknowledgments

We thank Veronique Delesalle for help with annotation review of our phage genomes, members of the Dieter Jahn Laboratory in the Institute of Microbiology at the Technische Universität Braunschweig especially Simone Virus for hosting DWB. Students in the SEA-PHAGES courses at IWU as well as students in the molecular biology course are thanked for their roles in phage isolation and initial annotation. Members of these classes are indicated in the S2 File. We also thank Dr. Barry Stein at the Electron Microscopy Center at the Indiana University Bloomington campus for electron microscopy.

## Author Contributions

**Conceptualization:** Richard M. Alvey, David W. Bollivar.

**Data curation:** Steven G. Cresawn, Richard M. Alvey, David W. Bollivar.

**Formal analysis:** Jackson Rapala, Brenda Miller, Maximiliano Garcia, Megan Dolan, Matthew Bockman, Daniel A. Russell, Rebecca A. Garlena, Alexander B. Westbye, Richard M. Alvey, David W. Bollivar.

**Writing – original draft:** David W. Bollivar.

**Writing – review & editing:** Megan Dolan, Mats Hansson, Daniel A. Russell, Steven G. Cresawn, Alexander B. Westbye, J. Thomas Beatty, Richard M. Alvey, David W. Bollivar.

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
