## [Decision Letter · Decision Letter 0]

3 Sep 2021

PONE-D-21-22171Genomic diversity of bacteriophages infecting Rhodobacter capsulatus and their relatedness to its gene transfer agent RcGTAPLOS ONE

Dear Dr. Alvey,

Thank you for submitting your manuscript to PLOS ONE. After careful consideration, we feel that it has merit but does not fully meet PLOS ONE’s publication criteria as it currently stands. Therefore, we invite you to submit a revised version of the manuscript that addresses the points raised during the review process.

Though lengthy, the reviewer's points consist mostly of clarifications and addition of details necessary to fully appreciate the findings of the research presented in the manuscript.

We look forward to receiving your revised manuscript.

Kind regards,

Gabriel Moreno-Hagelsieb

Academic Editor

PLOS ONE

“The research of JTB was supported by a grant (RGPIN 2018-03898) from the Canadian Natural Sciences and Engineering Research Council (NSERC). The research of DWB was supported by funds from the Illinois Wesleyan University Miner Linnaeus Sherff endowed professorship in Botany”

“The research of JTB was supported by a grant (RGPIN 2018-03898) from the Canadian Natural Sciences and Engineering Research Council (NSERC). The research of DWB was supported by funds from the Illinois Wesleyan University Miner Linnaeus Sherff endowed professorship in Botany. The funders had no role in study design, data collection and analysis, decision to publish, or preparation of the manuscript.”

Additional Editor Comments (if provided):

Reviewers' comments:

Reviewer's Responses to Questions

**Comments to the Author**

1. Is the manuscript technically sound, and do the data support the conclusions?

Reviewer #1: Yes

2. Has the statistical analysis been performed appropriately and rigorously? 

Reviewer #1: N/A

3. Have the authors made all data underlying the findings in their manuscript fully available?

Reviewer #1: Yes

4. Is the manuscript presented in an intelligible fashion and written in standard English?

Reviewer #1: Yes

5. Review Comments to the Author

Reviewer #1: In the study by Rapala and colleagues, the authors describe the isolation and characterisation of 26 phages infecting Rhodobacter capsulatus, a bacterial model for photosynthesis and nitrogen fixation for which phage discovery has been neglected for decades. The authors put particular emphasis on the genomic characterisation of the phage collection, placing the genomic sequences in the context of other R. capsulatus phages and the transfer agent RcGTA.

The study presents a detailed analysis of new phage genomes, increasing the known diversity for R. capsulatus and providing information valuable for phage researchers interested in this bacterial model. Yet, several passages of the manuscript lack information relevant to understanding the context or the methods behind the results presented, thus affecting the reproducibility of the work. Below, I have listed a series of points that I consider require addressing.

Major comments

Line (L) 26. The authors claim the report of 26 new bacteriophages; however, only 21 genome accessions are provided and from Table 3, the Results and Methods sections, it is not clear which phages are reported and sequenced in this study. The authors should clarify this in a revised version of the manuscript in all relevant sections (including the abstract) and Figure/Table legends.

L120-124. Indicate in the Table legend the number of virions measured. Likewise, as the table contains average values, include a value representing the level of variation amongst the measurements in parentheses.

L128. Why/how the potential host strains were chosen? Are they representative of the species diversity?

The methods lack critical information and precision, including

Versions and settings of the software used (even if they were run on default settings this has to be indicated)

Relevant cut-off values used to determine the sequence clusters and during the capsid and terminase BLASTp searches.

The bioinformatics strategy to predict phage life style and determining the type of genome end. The latter being particularly important to support genome completeness.

L178-180. The authors should consider repeating the search restricting the search set to viral sequences (e.g. Viruses (taxid:10239) or Caudovirales (taxid:28883)) in cases where no matches against phage genomes were found in the list of top 100 hits, and amend the relevant table and text if required.

L182. In table 4, add the % of coverage and sequence identity of the reported matches.

L190. “Very low numbers” is ambiguous, indicate the range of shared genes for singletons and in methods clarify what was the cut-off value to define them as such.

Figures 3 and 4 are redundant regarding the primary information they provide. I suggest the authors keeping Figure 4 and sending Figure 3 to supplementary material.

The manuscript contains an excessive number of genome comparisons as main figures, 7 in total (Figures 5-10 and 13). I encourage the authors to send these per-group comparisons to the supplementary material and compiling one main figure with the genome maps of the representatives of each group plus those of the singletons. Figure 13 displays relationships between some of the phages and RcGTA and can therefore stay as part of the main text.

The authors should ensure that all Figures are provided in the optimal resolution before publication, as the current ones do not allow close inspection when zooming in. This is especially relevant for the text and numbers embedded in the genome maps.

The authors report a diversity of GC content on the phage genomes sequenced. How do these different values compare to the GC content of the host genome? I recommend briefly discussing this topic in the relevant section of the manuscript.

L600. What do the authors mean by “It was not able to be further cultivated”? No new plaques were generated on a bacterial lawn from a previous plaque? Please clarify.

L649. Clarify in the figure legend what components and colours of the figure correspond to nucleotide or protein similarity and the minimum similarity value.

L634. It is quite interesting that the authors did not find many homologs between RcGTA and the phages reported. Has RcGTA been previously compared to other phages? Briefly discuss this in the relevant section (e.g. L684-685).

To ensure experiments reproducibility, it is essential that the authors provide information missing in the Methods section.

In the growth and Isolation sub-section: Add how long you grew the phage hosts in both liquid and solid cultures, including the number of propagation steps.

In the Lysogen testing sub-section: Indicate which host strains were used in these experiments; L729 - What PFU a “high titer lysate” corresponds to?; L733 and L736 - "After several days of incubation" and “several days until sufficiently grown” is ambiguous, state what the incubation period was and what sufficiently grown means (e.g. OD).

In the Sequencing and annotation sub-section: L758 - Clarify the origin and titer of the phage sample; L767 - Was the Illumina library paired-end? Clarify. Indicate the range of genome coverage achieved.

Minor comments

Considering that R. capsulatus is a model for photosynthesis, did the authors search for phage-encoded genes potentially involved in this or other cellular processes important for the host? Such genes have been previously described in phages; hence, a brief sentence in the discussion would be of value to the reader interested in this topic.

L101. Remove “gross”

L116-118. In Figure 1, it can be implied that the phage cluster is indicated above the phage name, but it would be clearer if this is indicated in the figure legend.

L131. Plaque morphology can be affected by several factors, and similar morphology does not guarantee sequence similarity. Thus, I recommend the authors focus this sentence on the patterns of plaque formation observed.

L142. If the authors are referring to plaques that are not clear, “turbid” is a more standard term than “cloudy”.

L157. Colour shades in the table are different from those used in figures. Overall, I consider the colours in the table unnecessary as the phage cluster is indicated.

L195. In Figure 2, it would be useful to mark (e.g. with an asterisk) which genomes were available prior to this study to easily distinguish them from the ones reported in the paper.

L220. Add the reference for Gepard in the figure legend.

L226. What does DSMZ stand for?

L231. Remove “phylogenomic” and clarify that the comparison was made at the nucleotide level (BLASTn).

Indicate the minimum level of pairwise sequence similarity in the legend of figures displaying comparative genome maps.

L285. Add a note indicating what “nkf” stands for.

L473. Replace “lysogens” for “prophages”

L521. “Myoviridae” shouldn’t be underscored.

L625. Remove “:”

L630. In Figure 13, both tail proteins are annotated as minor tail proteins.

L645. Indicate similarity levels at nucleotide or protein level rather than stating “substantial similarity”.

L663. What does the RMA and DWB abbreviations mean?

L681. Add a reference for “a pattern commonly seen with studies of phages”

L688-690. What is the role of the “peptidase”?

L704-706. Add an appropriate reference to this statement.

L712. Clarify how many phages reported in the study were isolated by the students.

L772-775. This paragraph should be moved into the Genome comparisons section.

6. PLOS authors have the option to publish the peer review history of their article (what does this mean?). If published, this will include your full peer review and any attached files.

Reviewer #1: No

---

## [Author Response · Author response to Decision Letter 0]

18 Oct 2021

The authors would like to thank the reviewers and editor for reading the manuscript and providing helpful comments and suggestions. Each item has been addressed below and highlighted in yellow. 

Adjustments to both the title page and main body of the document have been made to better adhere to these style requirements.

“The research of JTB was supported by a grant (RGPIN 2018-03898) from the Canadian Natural Sciences and Engineering Research Council (NSERC). The research of DWB was supported by funds from the Illinois Wesleyan University Miner Linnaeus Sherff endowed professorship in Botany”

“The research of JTB was supported by a grant (RGPIN 2018-03898) from the Canadian Natural Sciences and Engineering Research Council (NSERC). The research of DWB was supported by funds from the Illinois Wesleyan University Miner Linnaeus Sherff endowed professorship in Botany. The funders had no role in study design, data collection and analysis, decision to publish, or preparation of the manuscript.”

The funding information has been removed from the Acknowledgments Section.

All data for this manuscript is now publicly available.

The two instances of “data not shown” have been removed and the text has been modified for clarity.

The reference list is complete and correct. We do not believe any of the references have been retracted. 

Additional Editor Comments (if provided):

Reviewers' comments:

Reviewer's Responses to Questions

Comments to the Author

Reviewer #1: In the study by Rapala and colleagues, the authors describe the isolation and characterisation of 26 phages infecting Rhodobacter capsulatus, a bacterial model for photosynthesis and nitrogen fixation for which phage discovery has been neglected for decades. The authors put particular emphasis on the genomic characterisation of the phage collection, placing the genomic sequences in the context of other R. capsulatus phages and the transfer agent RcGTA.

The study presents a detailed analysis of new phage genomes, increasing the known diversity for R. capsulatus and providing information valuable for phage researchers interested in this bacterial model. Yet, several passages of the manuscript lack information relevant to understanding the context or the methods behind the results presented, thus affecting the reproducibility of the work. Below, I have listed a series of points that I consider require addressing.

Major comments

Line (L) 26. The authors claim the report of 26 new bacteriophages; however, only 21 genome accessions are provided and from Table 3, the Results and Methods sections, it is not clear which phages are reported and sequenced in this study. The authors should clarify this in a revised version of the manuscript in all relevant sections (including the abstract) and Figure/Table legends.

The correct number is 26. Since 5 of those were covered in a Genome Announcement it appears that those were already published somewhere else but Genome Announcements specifically state that publication with them does not preclude publication elsewhere. We have added the text “this publication” to the phage table to indicate that these 5 phages should be included in the 26 from IWU.

L120-124. Indicate in the Table legend the number of virions measured. Likewise, as the table contains average values, include a value representing the level of variation amongst the measurements in parentheses.

We have added the number of virions measured to the table and included standard deviation values. 

L128. Why/how the potential host strains were chosen? Are they representative of the species diversity?

This line was added to clarify how host strains were chosen:

 “These strains were chosen because they represent a spectrum of RcGTA interactions, in terms of either production or reception, and all except for Iona and St. Louis have a sequenced genome available”

The methods lack critical information and precision, including

Versions and settings of the software used (even if they were run on default settings this has to be indicated) 

Software versions and settings have been added to the text.

Relevant cut-off values used to determine the sequence clusters and during the capsid and terminase BLASTp searches.

The requested information has been added to the text

The bioinformatics strategy to predict phage life style and determining the type of genome end. The latter being particularly important to support genome completeness.

A description has been added to the methods section describing end determination and providing a reference for the method used.

L178-180. The authors should consider repeating the search restricting the search set to viral sequences (e.g. Viruses (taxid:10239) or Caudovirales (taxid:28883)) in cases where no matches against phage genomes were found in the list of top 100 hits, and amend the relevant table and text if required.

The queries were redone using the suggested restrictions and the resulting table has been edited along with the relevant text to reflect this alternative approach.

L182. In table 4, add the % of coverage and sequence identity of the reported matches.

The requested information has been added to the table.

L190. “Very low numbers” is ambiguous, indicate the range of shared genes for singletons and in methods clarify what was the cut-off value to define them as such.

The range was between 1 and 17 shared genes and has been indicated in the text along with a clarification of the cutoff value used by phamerator in its pham determination algorithm. 

Figures 3 and 4 are redundant regarding the primary information they provide. I suggest the authors keeping Figure 4 and sending Figure 3 to supplementary material.

Figure 3 has now been designated Supplementary figure S1 and the numbering of the other figures has been adjusted to accommodate this change.

The manuscript contains an excessive number of genome comparisons as main figures, 7 in total (Figures 5-10 and 13). I encourage the authors to send these per-group comparisons to the supplementary material and compiling one main figure with the genome maps of the representatives of each group plus those of the singletons. Figure 13 displays relationships between some of the phages and RcGTA and can therefore stay as part of the main text.

As per the reviewer’s suggestion, a new figure has been created (Figure 3) that compares representatives from each cluster and the two new singletons. Figures 5-12 were re-designated as figures S2-S9.

The authors should ensure that all Figures are provided in the optimal resolution before publication, as the current ones do not allow close inspection when zooming in. This is especially relevant for the text and numbers embedded in the genome maps.

Figures were re-worked and difficult-to-read small text was either resized or removed.

The authors report a diversity of GC content on the phage genomes sequenced. How do these different values compare to the GC content of the host genome? I recommend briefly discussing this topic in the relevant section of the manuscript.

This information is presented in the Genometrics section. All of the described phages have a GC content that is less than that of the host (66.5%).

L600. What do the authors mean by “It was not able to be further cultivated”? No new plaques were generated on a bacterial lawn from a previous plaque? Please clarify.

To clarify we added this language:

“Plaque formation on these two hosts only occurred with high titer (greater than 109 PFU/ml) lysate aliquots placed directly onto cells embedded in top agar. Further cultivation of this phage on either of these strains was not possible (Table 2).”

High concentrations of this phage could lyse this alternate host but it does not appear that the phage was able to use these cells for replication.

L649. Clarify in the figure legend what components and colours of the figure correspond to nucleotide or protein similarity and the minimum similarity value.

The figure legend has been updated as suggested.

L634. It is quite interesting that the authors did not find many homologs between RcGTA and the phages reported. Has RcGTA been previously compared to other phages? Briefly discuss this in the relevant section (e.g. L684-685).

We added a paragraph about this with references.

To ensure experiments reproducibility, it is essential that the authors provide information missing in the Methods section.

In the growth and Isolation sub-section: Add how long you grew the phage hosts in both liquid and solid cultures, including the number of propagation steps.

Cells were routinely cultivated for 2-3 days in both liquid and on solid media. Clarification was added to the text

In the Lysogen testing sub-section: Indicate which host strains were used in these experiments; L729 - What PFU a “high titer lysate” corresponds to?; 

The sentence was updated to indicate the strain used was YW1 and “high titer lysate” corresponds to a solution with >108pfu/ml.

L733 and L736 - "After several days of incubation" and “several days until sufficiently grown” is ambiguous, state what the incubation period was and what sufficiently grown means (e.g. OD).

The term “several days” was substituted with a more definitive 3-7 day time frame that more accurately reflects what was done for this experiment. The OD600 for that amount of time is typically between 0.8 - 1.0. This has been added to the text. 

In the Sequencing and annotation sub-section: L758 - Clarify the origin and titer of the phage

The sentence was edited to indicate that the lysate used for DNA preparation had greater than 108 pfu/ml and was harvested from a fully lysed sample of R. capsulatus on a plate.

 sample; L767 - Was the Illumina library paired-end? Clarify. Indicate the range of genome coverage achieved. 

The text has been updated to indicated that the libraries were paired-end and the ranges for the fold coverage have been added. Individual genome coverage results were also added to Table 3.

Minor comments

Considering that R. capsulatus is a model for photosynthesis, did the authors search for phage-encoded genes potentially involved in this or other cellular processes important for the host? Such genes have been previously described in phages; hence, a brief sentence in the discussion would be of value to the reader interested in this topic.

No evidence for photosynthesis related genes was found. A sentence has been added to the Discussion 

“It does not provide evidence for any photosynthesis related genes carried in the genome of these bacteriophages. “

L101. Remove “gross”

Done.

L116-118. In Figure 1, it can be implied that the phage cluster is indicated above the phage name, but it would be clearer if this is indicated in the figure legend.

Changed as suggested.

L131. Plaque morphology can be affected by several factors, and similar morphology does not guarantee sequence similarity. Thus, I recommend the authors focus this sentence on the patterns of plaque formation observed.

Changed as recommended.

L142. If the authors are referring to plaques that are not clear, “turbid” is a more standard term than “cloudy”.

Changed as suggested.

L157. Colour shades in the table are different from those used in figures. Overall, I consider the colours in the table unnecessary as the phage cluster is indicated.

Colors have been removed from the table.

L195. In Figure 2, it would be useful to mark (e.g. with an asterisk) which genomes were available prior to this study to easily distinguish them from the ones reported in the paper.

The figure has been changed as suggested and the figure legend updated to indicate what the asterisks mean.

L220. Add the reference for Gepard in the figure legend.

Done. Added citation to : Jan Krumsiek, Roland Arnold, Thomas Rattei, Gepard: a rapid and sensitive tool for creating dotplots on genome scale, Bioinformatics, Volume 23, Issue 8, 15 April 2007, Pages 1026–1028, https://doi.org/10.1093/bioinformatics/btm039

L226. What does DSMZ stand for?

DSMZ = Deutsche Sammlung von Mikroorganismen und Zellkulturen (German Collection of Microorganisms and Cell Cultures) 

The text has been adjusted to make this clear.

L231. Remove “phylogenomic” and clarify that the comparison was made at the nucleotide level (BLASTn).

Text changed as requested.

Indicate the minimum level of pairwise sequence similarity in the legend of figures displaying comparative genome maps.

The minimum level of pairwise sequence similarity is provided in the legend for figure S2. The figure legends for the subsequent similar figures refer back to this figure legend.

L285. Add a note indicating what “nkf” stands for.

Done.

L473. Replace “lysogens” for “prophages”

Done.

L521. “Myoviridae” shouldn’t be underscored.

Done.

L625. Remove “:”

Done.

L630. In Figure 13, both tail proteins are annotated as minor tail proteins.

We have adjusted the figure to call one “distal tail protein” and the other “hub” as these genes have been recently annotated.

L645. Indicate similarity levels at nucleotide or protein level rather than stating “substantial similarity”.

The sentence has been changed to indicate the threshold used by Phamerator to make these designations.

L663. What does the RMA and DWB abbreviations mean?

We have changed this sentence to now read:

“The sequencing of phage genomes combined with the integration of phage discovery into the biology courses of authors RMA and DWB led to development of the collection of sequenced R. capsulatus phages described here.”

L681. Add a reference for “a pattern commonly seen with studies of phages”

Added a reference to this paper:

Hatfull GF, Hendrix RW. Bacteriophages and their genomes. Curr Opin Virol. 2011 Oct;1(4):298-303. doi: 10.1016/j.coviro.2011.06.009. PMID: 22034588; PMCID: PMC3199584.

L688-690. What is the role of the “peptidase”?

The text has been updated to include the role of this protein.

L704-706. Add an appropriate reference to this statement.

We have adjusted this sentence to now read:

“In addition, it provides suggestive evidence that phages that infect members of the Rhodobacteraceae may move between different host species, including between marine and freshwater environments. “

L712. Clarify how many phages reported in the study were isolated by the students.

The text has been updated to indicate that 15 of the 26 new phages reported here were isolated by students in the phage hunting class.

L772-775. This paragraph should be moved into the Genome comparisons section.

The paragraph was moved into the designated section as requested with minor adjustments to the wording.

---

## [Editor Report · Decision Letter 1]

2 Nov 2021

Genomic diversity of bacteriophages infecting Rhodobacter capsulatus and their relatedness to its gene transfer agent RcGTA

PONE-D-21-22171R1

Dear Dr. Alvey,

We’re pleased to inform you that your manuscript has been judged scientifically suitable for publication and will be formally accepted for publication once it meets all outstanding technical requirements.

Kind regards,

Gabriel Moreno-Hagelsieb

Academic Editor

PLOS ONE
---

## [Editor Report · Acceptance letter]

8 Nov 2021

PONE-D-21-22171R1 

Genomic diversity of bacteriophages infecting *Rhodobacter capsulatus* and their relatedness to its gene transfer agent RcGTA 

Dear Dr. Alvey:

I'm pleased to inform you that your manuscript has been deemed suitable for publication in PLOS ONE. Congratulations! Your manuscript is now with our production department. 

Kind regards, 

on behalf of

Prof. Gabriel Moreno-Hagelsieb 

Academic Editor

PLOS ONE